# Analysis of the Microstructure Development of Nb-Microalloyed Steel during Rolling on a Heavy-Section Mill

**DOI:** 10.3390/ma16010288

**Published:** 2022-12-28

**Authors:** Michal Sauer, Richard Fabík, Ivo Schindler, Petr Kawulok, Petr Opěla, Rostislav Kawulok, Vlastimil Vodárek, Stanislav Rusz

**Affiliations:** 1Liberty Ostrava, a.s., Vratimovská 689/117, 71900 Ostrava, Czech Republic; 2Faculty of Materials Science and Technology, VŠB-Technical University of Ostrava, 17., 70800 Ostrava, Czech Republic

**Keywords:** static recrystallization, strain-induced precipitation, microstructure evolution, PSCT

## Abstract

It is not realistic to optimize the roll pass design of profile rolling mills, which typically roll hundreds of profiles, using physical modelling or operational rolling. The use of reliable models of microstructure evolution is preferable here. Based on the mathematical equations describing the microstructure evolution during hot rolling, a modified microstructure evolution model was presented that better accounts for the influence of strain-induced precipitation (SIP) on the kinetics of static recrystallization. The time required for half of the structure to soften, *t*_0.5_, by static recrystallization was calculated separately for both situations in which strain-induced precipitation occurred or did not occur. On this basis, the resulting model was more sensitive to the description of grain coarsening in the high-rolling-temperature region, which is a consequence of the rapid progress of static recrystallization and the larger interpass times during rolling on cross-country and continuous mills. The modified model was verified using a plain strain compression test (PSCT) simulation of rolling a 100-mm-diameter round bar performed on the Hydrawedge II hot deformation simulator (HDS-20). Four variants of simulations were performed, differing in the rolling temperature in the last four passes. For comparison with the outputs of the modified model, an analysis of the austenite grain size after rolling was performed using optical metallography. For indirect comparison with the model outputs, the SIP initiation time was determined based on the NbX precipitate size distribution obtained by TEM. Using the PSCT and the outputs from the modified microstructure evolution model, it was found that during conventional rolling, strain-induced precipitation occurs after the last pass and thus does not affect the austenite grain size. By lowering the rolling temperature, it was possible to reduce the grain size by up to 56 μm, while increasing the mean flow stress by a maximum of 74%. The resulting grain size for all four modes was consistent with the operating results.

## 1. Introduction

Thermomechanical steel processing, also referred to in the literature as controlled forming, aims to achieve the desired microstructure, i.e., the desired mechanical and physical properties, of the final product. This is primarily achieved by refining the final ferritic grain, which is transformed from deformed austenite. Controlled forming is, therefore, possible in order to achieve the desired properties of the product, with a significant reduction in the alloying content elements and heat treatment costs compared to conventionally rolled material [1,2]. Controlled forming has been extensively described in the literature [2,3,4,5]. Therefore, the following text briefly describes three selected types of controlled forming using mainly retarding austenite recrystallization due to a combination of microalloying elements in solid solution and interactions between recrystallization and strain-induced precipitation (SIP).

Recrystallization-controlled rolling is commonly used for rolling thick plates and thin-walled seamless tubes, where the rolling forces are close to the limit value of rolling mills [6]. The forming process is carried out at high temperatures (above 950 °C), and complete static recrystallization (SRX) can occur during the interpass times. For this purpose, steel microalloying elements, i.e., Ti and V, are added to the steel to allow complete static recrystallization (SRX) between passes and simultaneously prevent grain growth after recrystallization before the following pass. The addition of Ti leads to the precipitation (SIP) of TiN during the continuous casting of the billets, which then prevents extensive grain growth after recrystallization. The addition of V leads to the precipitation of VN in the ferritic region, which leads to hardening of the steel. The recrystallization-controlled forming process should be associated with high cooling rates to achieve a fine ferritic grain after transformation from austenite. This method can achieve a ferritic grain size of about 7 to 10 μm [2,3,4,5,6,7].

Conventional controlled rolling aims to produce flattened austenitic grains due to plastic deformation without any softening process, leading to increased nucleation sites for transforming from austenite to ferrite. This process then leads to the formation of fine ferritic grains approximately 5 to 8 μm in size. Conventional controlled rolling generally involves high heating temperatures to achieve the complete transition of the microalloying elements, i.e., Nb and V, into a solid solution. During the rolling process, which takes place below zero recrystallization temperature, strain-induced precipitation of Nb(C, N) occurs, causing complete suppression of recrystallization between each pass [2,3,4,5,6,7].

Dynamic recrystallization-controlled rolling involves the initiation of dynamic recrystallization in one or more passes during the rolling process. It is characteristic for rolling wires and bars on continuous rolling lines but also rolling strips and seamless tubes [7]. This can be achieved by applying a large amount of deformation in one pass or accumulating strain in several individual passes. In the case of both methods, a critical strain is required to initiate dynamic recrystallization. The final ferrite grain size can reach 1 to 2 μm [2,3,4,5,6,7].

Many authors [8,9,10,11] have used microstructure evolution models to design controlled forming regimes. Usually, however, these are models created for strip rolling on the final sequence of (semi)continuous mills at low rolling temperatures, where the SIP effect has already caused, or will soon cause, a slowing down (in the case of continuous mills with short interpass times, a practical stop) of recrystallization. Our experience shows that such models do not give reliable results when simulating rolling of long products at higher temperatures. This is due to the use of a single equation to calculate *t*_0.5_ for both situations in which SIP did or did not take place.

The purpose of this study is to propose and verify a new model for microstructure evolution that will be more accurate in both situations.

## 2. Microstructure Evolution Model

In recent decades, considerable attention has been paid to developing mathematical models that predict the microstructure evolution of hot-rolled products [1,2,3,4,5,6,7,8,9,10,11,12,13,14,15,16,17,18,19,20,21,22,23]. A significant advantage of these models is the possibility to optimize the rolling conditions and thus obtain a good combination of thermomechanical parameters to achieve optimal mechanical properties of the final product. A weakness of most mathematical models described in the literature is the weak connection between the processes of precipitation and recrystallization. This, then, especially in the case of a cross-country rolling mill (higher interpass time) at higher temperatures, leads to unrealistically fast static recrystallization (SRX) and thus to enormous coarsening of the austenitic grain. Therefore, this work aims to modify the mathematical models predicting austenitic grain evolution during hot rolling to better account for the relationship between recrystallization and precipitation.

### Description of the Microstructure Development Model Modification

In the literature [24,25,26,27,28,29,30], one can encounter two-step curves describing the growth of the fraction of the softened structure as a function of the thermodynamic conditions of the forming process, i.e., temperature, strain, strain rate and grain size, during the interpass time. The most complete data set represents the work of authors around Medina [27,28,29,30,31,32,33,34,35,36,37]. The data of these authors were used in the modification of our model. Figure 1 is an example of the two-stage softening curves [37] for steel, with the chemical composition listed in Table 1 for N8 steel.

These curves generally determine the time to soften half of the structure, *t*_0.5_, that occurs in the JMAK equation. This is the imaginary intersection of the value of *X* = 0.5 with these curves. In the case of simple S-curves, this is their inflection point. In the case of two-degree curves, this rule no longer applies. The values of *t*_0.5_ thus obtained are used to develop an equation that respects the effect of precipitation on the kinetics of recrystallization. The disadvantage, however, is that the equation thus developed leads to a shift of *t*_0.5_ to high times in cases where the conditions for initiating strain-induced precipitation (SIP) have not yet been fulfilled during the previous rolling. Moreover, in the temperature region of the nose of the curve of the onset of precipitation, the values of *t*_0.5_ in the equations thus developed are biased because they do not consider the step effect of the precipitation on the kinetics of recrystallization. We attempted to use published data for steels N3, N4 and N8 to develop new equations to calculate *t*_0.5_ separately for the situation in which SIP takes place and separately for the situation in which SIP does not occur at all.

The experimental data were fitted with Avrami S-curves so that the sum of the squares of the deviations of the measured values from these curves was minimal. In the case of two-degree curves, two S-curves were used separately for the two degrees of the curves. Data occurring in the region of a constant proportion of the softened structure were not counted (see Figure 2).

In this way, we first obtained the values of the coefficient *n* in the Avrami equation (see Table 2). First, we needed to check whether the coefficient *n* is a function of temperature, as some authors state, or whether it is affected by other influences.

The negligible effect of temperature on the coefficient *n* is illustrated in Figure 3. The effect of the Nb content and strain size was similar.

Finally, the value of *n* was determined as the average for the curves without SIP (*n* = 0.823) and with SIP (*n* = 0.484). These values were used to correct the S-curves in the plots similar to Figure 2 (separately for each of the 3 steels and for the 2 strain values of 0.2 and 0.35). By varying the value of the *n* coefficient in each curve, the curves were shifted towards the optimum based on the least squares method. Therefore, the position of the curves was changed again so that the minimum sum of the squares of the deviations of the measured points from the S-curves was again achieved. From the curves thus generated, a *t*_0.5_ value was determined for each curve separately (see Table 3).

The data from Table 3 were plotted against the reciprocal thermodynamic temperature (see Figure 4). The activation energy values (exponent multiplied by the molar gas constant; see Table 4) and the value of the *K* constant were determined using exponential regression.

The graph in Figure 5 shows the activation energy of the studied steels as a function of the Nb content and strain value in both the non-SIP and SIP conditions. In both cases, there seemed to be a statistically significant dependence, so the activation energy values were determined by multilinear regression as a function of the Nb content and strain in the form:

Without SIP:(1)Q=1089469·%Nb−121212·e+248346

With SIP:(2)Q=418324·%Nb−416137·e+496325

A comparison of the measured and calculated values according to Equations (1) and (2) is made in Figure 6.

The equations obtained to describe *t*_0.5_ had the following form:

Without SIP:(3)$$t_{0.5}=\exp\left(6.036\cdot e-80.87\cdot \%Nb-19.83\right)\;\;\cdot \exp\left(\frac{1089469\cdot \%Nb-121212\cdot e+248346}{R\cdot T}\right)$$

With SIP:(4)$$t_{0.5}=\exp\left(32.235\cdot e-329.6\cdot \%Nb-43.24\right)\;\;\cdot \exp\left(\frac{4184324\cdot \%Nb-416137\cdot e+496325}{R\cdot T}\right)$$

The comparison of the measured values according to Equations (3) and (4) is shown in the plot in Figure 7. The conformity between the measured and calculated values for *t*_0.5_ without SIP is good, but in the case of *t*_0.5_ with SIP, the scatter of the data around the mean line is visibly worse. The arrangement of the data into layers by steel number can be seen here (data labels in the plot in Figure 7b). All *t*_0.5_ values are below the mean line for N3 steel, while the opposite is true for N8 steel. Apart from the Nb content, the steels used differ mainly in the Mn and Si content ratio, significantly affecting Nb precipitation. This effect is more pronounced in steels with a higher Nb content because the solubility of Nb in steel decreases with an increasing Mn/Si ratio. Therefore, Equation (4) was modified to the following form:(5)$$t_{0.5}=\exp\left[1.37\cdot 10^{-4}\cdot {\left(\frac{\%Mn}{\%Si}\right)}^{5.1282}\right]\cdot \exp\left(32.235\cdot e-329.6\cdot \%Nb-43.24\right)\;\cdot \exp\left(\frac{4184324\cdot \%Nb-436137\cdot e+496325}{R\cdot T}\right)$$

Using this correction factor, the scatter of values around the mean line was minimized (Figure 8). If the data were further refined, more measured softening curves’ data would be required for the variant with SIP.

Since in the experiment from which our data were taken, the grain size ranged from 140 to 210 μm, it was necessary to add a term to the equations to describe *t*_0.5_ to account for the change in grain size during actual rolling. The effect of grain size *D* was included in previous equations in the following form:(6)$$t_{0.5\left(D\right)}=\frac{t_{0.5\left(D\right)}}{D^{\prime }}$$
where *D*′ is the average value of the grain size used in the experiment (180 μm).

To use the equations obtained to describe *t*_0.5_ in the microstructure evolution model, it is necessary to solve the determination of *t*_0.5_ in the pass in which the SIP occurs. In this pass, it is assumed that *t_ip_* > *t_ps_*. The following two situations can occur (see Figure 9). The fraction of the softened structure at time *t_ip_* for a curve with SIP is

−greater than the proportion of the softened structure at time *t_ps_* for the curve without SIP (*t_ps_*_1_ and *X_tps_*_1_ in Figure 9);−less than the proportion of the softened structure at time *t_ps_* for the curve without SIP (*t_ps_*_2_ and *X_tps_*_2_ in Figure 9).

In the first case, the model will calculate the X values from the SIP curve and vice versa in the second case.

The last point of the model modification is the modification of the parameters of the Hodgson equation [37] for the grain size calculation after static recrystallization. The data obtained from the PSCT (see the next section) showed that our HSLA steel has a significantly coarser grain. The modified equation has the following form:(7)$$d_{SRX}=450\cdot d_{0}^{0.4}\cdot e^{-0.5}\cdot exp\left(\frac{-45000}{R\cdot T}\right)$$

## 3. Plain Strain Compression Test (PSCT)

### 3.1. Experiment Description

HSLA steel (S355J2), whose chemical composition is shown in Table 5, was selected to verify the modified microstructure evolution model. A total of 5 cubic specimens of 20 × 20 × 20 mm were prepared from the 180 × 180 mm continuously cast block specimen for the laboratory experiments to determine the carbonitride dissolution temperature during heating and 4 specimens of 10 × 15 × 20 mm for the plain strain compression test (PSCT).

The aim of the laboratory simulations was:

(1) to determine the temperature for complete dissolution of carbonitrides during heating;

(2) PSCT simulation of rolling a round bar of 100 mm diameter on a cross-country rolling mill at Liberty Ostrava a.s.

(3) analysis of the microstructure of samples after PSCT by transmission electron microscopy (TEM).

To determine the amount of Nb in the solid solution γ after heating, a simple quenching and tempering experiment was performed on samples heated to different heating temperatures. A total of 5 samples were individually heated in an electric resistance furnace to a temperature of 1100–1150–1200–1220–1240 °C for 30 min. After the hold time, the samples were quenched in water with ice and then tempered at 575 °C for 60 min. After the specimens were cut, ground and polished, an average hardness of HBW 30 was determined in the central region of the specimens, based on 5 imprints each time. Based on the results shown in Figure 10, the heating temperature required for complete dissolution of the precipitates was determined to be 1200 °C.

In the plane strain compression tests performed on the Hydrawedge II hot deformation simulator HDS-20, anvils with a working width of 5 mm were pressed into the test specimens with dimensions of 10 × 15 × 20 mm. A total of 4 simulations were performed with a uniform heating temperature of 1200 °C with a 15 min hold at this temperature. The true strain, calculated based on the elongation factor, was converted using the control program with a conversion factor of 1.155 to strain intensity values that respected the law of volume conservation, along with dimensional changes in all 3 directions. In total, 7 passes were simulated in each variant at deformation temperature *T*, strain rate *ė* and interpass times *t_ip_*. A simplified schematic representation of the PSCTs is shown in Figure 11.

### 3.2. Microstructural Analysis

The microstructure of samples V1–V4 (corresponding to PSCT simulation variants 1–4) was preferentially examined in the centre of the deformed region in terms of its width and height.

Figure 12 documents the microstructure in the deformed part of sample V1, along with the marked areas of detailed microstructural analysis.

In general, the microstructure in the undeformed part of all samples was comparable: allotriomorphic ferrite along the boundaries of the original austenitic grains. The remaining microstructure was predominantly a mixture of Widmanstätten ferrite and a minor perlitic phase. The presence of a small proportion of bainite could not be dismissed (see Figure 13). Due to allotriomorphic ferrite, which nucleates at the original austenitic grain boundaries and continues to grow preferentially at these boundaries, we can estimate the austenitic grain size after heating (see the red original austenite boundary in Figure 13). Using image analysis in ImageJ software, the size of the largest original austenitic grains was determined (the position of the cutting plane strongly influences the measured size, we can assume that the structure is homogeneous in terms of grain size and the largest grains are those that are split by the cutting plane close to their centre). The results of the measurements are summarized in Table 6.

The so-called forging cross was clearly visible in samples V3 and V4. In the centre of the forging cross, the microstructure of these samples was a fine-grained mixture of ferrite and pearlite—Figure 14 and Figure 16. In samples V1 and V2, without a distinct forging cross, the microstructure in the centre of the cross-section of the samples was a mixture of allotriomorphic ferrite, Widmanstätten ferrite and pearlite—Figure 15 and Figure 16. Due to the presence of allotriomorphic ferrite, it is again possible to estimate the austenitic grain size before the onset of phase transformation using image analysis. The results of the measurements are summarized in Table 7, in which the average microhardness values are also given for comparison.

### 3.3. Precipitate Analysis

TEM studies were performed using carbon extraction replicas. Sample masking was used in the preparation of the replicas in order to obtain slides only from the centre of the deformed part of the samples. When studying the replicas in TEM, areas where the microstructure was mainly ferrite and pearlite were preferred. Documentation of precipitate particles was performed only in ferritic grains. Minority phases on the replicas were identified using energy-dispersive X-ray spectroscopy (EDX). In all samples, only cementite particles (part of the pearlitic component, possibly bainite, whose presence cannot be excluded, although most of the austenite decay products morphologically corresponded to Widmanstätten ferrite) and NbX phase particles were detected. EDX showed that small amounts of titanium, chromium and niobium were present in the NbX particles (Table 8). The presence of small amounts of iron in the EDX spectra probably represents an artefact associated with the preparation of the slides. No clear differences were found between the chemical composition of coarser and fine NbX particles. Given the chemical composition of HSLA steel, it can be assumed that the particles are carbides rather than carbonitrides.

In the studied samples, we could identify a total of three different sizes of NbX particles. In the variants, samples V1 and V2, only coarse particles with a size above 80 nm were present (see Figure 17. In samples V3 and V4, besides coarse particles, there were also fine particles with a size of around 60 nm and even fine particles below 20 nm (see Figure 17). In samples V3 and V4, it could be seen that the fine particles were arranged in rows, suggesting that they are precipitates formed during deformation when slip bands became the sites for the formation of the nuclei. If no further deformation occurs, the precipitates coarsen. If further deformation occurs, new slip bands and thus new sites for NbX nuclei appear, as the solid solution is depleted of Nb and C around the original slip bands. This would then cause the appearance of 2 peaks on the histogram in the region corresponding to the strain-induced precipitates. A detailed analysis of the occurrence of particles is made in the following section.

### 3.4. Simulation of PSCTs Using a Modified Microstructure Evolution Model

Using a modified microstructure evolution model, microstructure evolution was simulated for all 4 variants of PSCT rolling simulation. The grain size evolution for all PSCT variants is shown in Figure 18. For variant 1, which represents conventional rolling, the gradual grain refinement was due to fully completed static recrystallization after almost every pass. Figure 18 shows undesirable grain coarsening occurred after the 5th and 7th passes. During cooling after rolling, at 1009 °C and a time of *t_ip_*_,7_ = 49.44 s, SIP occurred. The time available for recrystallization or grain coarsening after the last pass of *t_ip_*_,7_, which represents the x-intercept of the length of the cooling curve before its intersection with the precipitation onset curve (see Figure 19a), was calculated using the following Equation (8):(8)$$t_{ip,7}=t_{PPT}-t_{ps,7}\cdot \overset{8}{\underset{i=1}{\sum }}\frac{t_{ip}}{t_{ps}}$$
where *t_SIP_* is the time of onset of precipitation during cooling (intersection of curves in Figure 19a).

Thus, the onset of precipitation in the case of variant 1 stops the grain coarsening process, and the resulting grain size before phase transformation was around 96 μm, which fit exactly the measured grain size from PSCT (96 μm).

The grain size evolution for variant 2 was almost identical to variant 1 until the last pass. Due to the faster onset of precipitation after rolling (*t_ip_*_,7_ = 10.0 s; see the x-projection of the length of the cooling curve before it intersects with the curve of the onset of precipitation in Figure 19b), there was no termination of static recrystallization (*X* = 0.42). Thus, the structure contained original unrecrystallized grains about 73 μm in size (58% of them) elongated in the direction of deformation before the phase transformation, which corresponds well to the measured grain size from the PSCT (72 μm) and recrystallized equal grains 48 μm in size. The residual strain before phase transformation of the austenite was about 0.13.

For variant 3, there was a noticeable decrease in grain size due to the reduced temperature before the last three passes. In the 6th pass, strain-induced precipitation (SIP) stopped the ongoing static recrystallization (*X* = 0.61), thus reducing the grain size to 40 μm, which corresponds fairly well to the measured grain size from the PSCT. In the last pass, no static recrystallization occurred despite the accumulated strain from the previous pass; thus, the grain size did not change. Thus, a relatively large, elongated grain (residual strain 0.6) with an area corresponding to an equal grain of 40 μm diameter was present in the structure before phase transformation.

In variant 4, compared to variant 3, SIP already occurred in the 5th pass. The SIP stopped the ongoing static recrystallization (*X* = 0.85), and the grain size after this pass dropped to 63 μm (the smallest of all variants at this rolling stage). However, in the subsequent passes, static recrystallization did not occur, so the grain size did not change. Only the rate of elongation due to the accumulated strain of 0.76 increased.

The mean flow stress (MFS) values predicted by the model were compared with the MFS values calculated from the measured forming forces during the PSCT according to the Formula (9).
(9)$$\sigma =0.866\cdot \frac{F}{l_{s}\cdot b_{0}}$$
where *F* (N) is the measured force, *l_s_* (mm) is the length of the contact area between the sample and the anvil (in our case *l_s_* = 5 mm) and *b*_0_ (mm) is the width of the tested sample (in our case *b*_0_ = 20 mm).

According to the modified Misaka equation, the MFS values were 25 to 100 Mpa lower than the MFS values calculated from the measured forces. Therefore, we built our model in the following form for further simulation:(10)$$MFS=6\cdot e^{0.179}\cdot {\dot{e}}^{0.081}\cdot \exp\left(\frac{4263}{T}\right)$$

A comparison of all the MFS values calculated from the measured forces according to Equation (10) is plotted in Figure 20. The dependence of MFS on the temperature and strain rate calculated according to Equation (10) is shown in Figure 21.

The dependence of the MFS on the reciprocal temperature for all 4 rolling variations is shown in Figure 22. From the 5th pass onwards, we saw an MFS increase due to a rolling temperature decrease. The high MFS value in the 6th pass was due to the largest partial strain increase due to the cumulative strain from the previous passes (for variants 1 to 3, we had a cumulative strain in the 6th pass of about 0.4; for variant 4, it was already 0.56 due to the SIP started in the 5th pass). For variants 1 and 2, the increase in the MFS in the last pass was relatively small compared to the increase for variants V3 and V4, where static recrystallization did not occur due to the ongoing SIP, and we had cumulative strain values of 0.6 and 0.76 in the last pass for variants 3 and 4, respectively. Using a rolling mode that induces SIP while rolling is still in progress, we estimated an MFS increase of at least 50%. In our simulation, this was 53 and 74% for variants 3 and 4, respectively. Similar percentage increases can be expected for roll-separating forces and torques during real rolling.

### 3.5. Analysis of Precipitate Size Distribution

Simulations of each variant using the modified model were now used to clarify the occurrence of precipitates of different sizes in the PSCT samples for all variants (see Figure 17). All available images from the previous section were subjected to image analysis in ImageJ software. The results of this analysis are presented for all variants using histograms for particle diameter values (calculated from the particle area). Using a single histogram to represent the frequency of occurrence of precipitates of different sizes over the whole observed range from 1 nm to 250 nm is misleading, as fine precipitates are much more abundant than coarse ones. Therefore, we divided the data into two groups (group 1: 1 nm to 40 nm; group 2: 40 nm to 250 nm) using the absolute frequency for fine precipitates and the relative area of coarse precipitates in the structure (area of all precipitates in a given class divided by the total area of all precipitates). Histograms depicting the size distribution of fine precipitates are shown in Figure 23. 

For variants 1 and 2, there were practically no precipitates with a size below 40 nm in the structure, so we did not show histograms here. For variant 3, the histogram had a regular bell shape, with a mean value of 9 nm and a standard deviation of 5 nm. For variant 4, we saw a classical two-peaked histogram, which indicates that the data came from two independent sets. The first had a mean value of 4 nm and a standard deviation of 3 nm, while the second had a mean value of 15 nm and a standard deviation of 4 nm.

The histograms in Figure 24 were influenced by the significantly lower frequency of measured data (precipitates below 40 nm were measured in total for variants 3 and 4 over 6300, but those above 40 nm were only 295 for all variants). Here, we also considered two peaks for all variants. One (represented in all graphs by a Gaussian curve with a mean value of 172 nm) corresponded to precipitates not dissolved when heated to the forming temperature. The mean values were calculated from all the variants together because, according to the results of other authors (Vervynckt et al. [25]), it can be assumed that the actual forming regime does not influence the size of the undissolved precipitates. The second peak of the histogram was then at a different mean value of precipitate size for each variant, with the mean value shifting towards smaller values as the number of variants increased.

Now that we know the precipitation history by simulating all the modified microstructure evolution model variants, we can analyze the precipitate size distribution in HSLA steel in detail.

For variants 1 and 2, we had two groups of precipitates:−undissolved precipitates with a mean value of 172 nm;−precipitates formed during cooling after rolling with mean values for V1 and V2 of 110 and 105 nm, respectively. The lower value for V2 is due to the lower precipitation temperature (1009 °C for V1 vs. 989 °C for V2).

In variant V3, we had 3 groups of precipitates: −undissolved precipitates with a mean value of 172 nm; −precipitates formed during cooling after rolling with a mean value of 64 nm (precipitation temperature 944 °C, solid solution partially depleted of Nb due to previous precipitation);−precipitates formed during precipitation in the pause between the 6th and 7th passes; these have a mean size of only 9 nm due to the limited time for precipitation (24 s).


For variant V4, we had even 4 groups of precipitates:−undissolved precipitates with a mean value of 172 nm; −precipitates formed during cooling after rolling with a mean value of 56 nm (precipitation temperature 914 °C, solid solution partially depleted of Nb due to previous precipitation);−precipitates formed during precipitation in the pause between the 6th and 7th passes; these have a mean size of 15 nm due to the limited time for precipitation (32 s);−precipitates formed during precipitation in the pause between the 5th and 6th passes; these have an average size of only 4 nm due to the limited time for precipitation (19 s).

Based on these values of precipitation parameters and the corresponding values of precipitate size, it was possible to develop Equation (11) describing the effect of temperature and precipitation time on the size of NbX precipitates:(11)$$D_{PPT}=616.9\cdot t^{0.742}\cdot \exp\left(\frac{-8195.8}{T}\right)$$

The graph in Figure 25 documents the accuracy of Equation (11). The dependence of precipitate size on time and precipitation temperature calculated by Equation (11) is shown in Figure 26. This equation can now be included in the modified microstructure evolution model.

## 4. Discussion

Based on the modification of the mathematical model, simulations of the microstructure evolution during the rolling of a round bar from HSLA steel with a diameter of 100 mm were performed, including a comparison with the operational results from a heavy-section mill.

Four situations corresponding to the plain strain compression test were modelled. In the case of the first scenario corresponding to conventional rolling, SIP occurs only after the last pass, which does not affect grain size. The predicted grain size is around 96 μm, which corresponds well to the operational results. The evolution of austenitic grain size is almost identical for variants 1 and 2. Due to the more rapid onset of SIP after rolling, SRX after the last pass is not terminated, and the structure contains original recrystallized grains with a size of about 73 μm elongated in the deformation direction and recrystallized equiaxed of about 48 μm before phase transformation. In the third variant, a noticeable decrease in austenitic grain size can be observed due to the reduced temperature before the last three passes. In the sixth pass, SIP occurs, which stops the ongoing SRX, and thus, the grain size reduces to about 40 μm. In the last pass, despite the accumulated strain from the previous pass, no SRX occurs; therefore, the grain size does not change. Thus, in the structure before phase transformation, a relatively large amount of elongated austenitic grain with a residual strain of 0.6 is present with an area corresponding to an equiaxed grain of 40 μm diameter. Compared to variant 4, there is a suspension of SRX due to SIP in the fifth pass. The grain size in this pass drops to 63 μm, which is the smallest of all simulated variants at this rolling stage. In the following passes, static recrystallization does not occur anymore, so the resulting austenitic grain size does not change. Only the elongation rate due to the accumulated strain of 0.76 increases. All simulation results thus correspond well with the operational results. From the MFS point of view, we can observe an increase in the values from the fifth pass onwards due to the decreasing rolling temperature. The accumulated strain due to the largest partial strain increases the high MFS value in the sixth pass from the previous. For variants 1 and 2, the increase in the last pass is relatively small compared to the increase in variants 3 and 4. Using a rolling mode that induces SIP while rolling, we estimate an increase in the MFS of at least 50%. In the case of our simulations, this was 53% for variant 3 and even 74% for variant 4. Similar percentage increases can be expected for roll-separating forces and torques during real rolling. 

Simulations of the different variants were used to clarify the occurrence of precipitates of different sizes, which showed that up to four types of precipitates are present in the investigated samples. These are precipitates that do not dissolve during heating and precipitates that form during rolling between passes, or precipitates that form during cooling.

## 5. Conclusions

This paper showed a modification of the mathematical model of microstructure evolution that reflects the effect of SIP on SRX kinetics in comparison with existing models. The model showed good agreement with the PSCT simulation results of rolling a 100-mm-diameter round bar of HSLA steel.

Currently, the model is used at Liberty Ostrava to optimize rolling temperatures when rolling round and flat bars and I, V and L profiles on a heavy-section mill (continuous cross-country-type mill). Its verification in rolling bars and profiles on medium- and fine-section mills (continuous mill) is under preparation.

## Figures and Tables

**Figure 1 materials-16-00288-f001:**
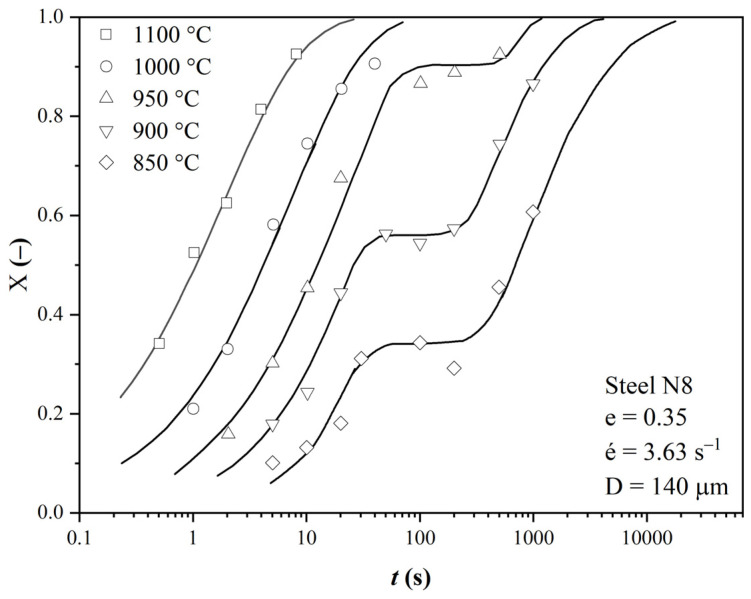
Variation in the recrystallized fraction (*X_a_*) with time (*t*), Reprinted from Ref. [14].

**Figure 2 materials-16-00288-f002:**
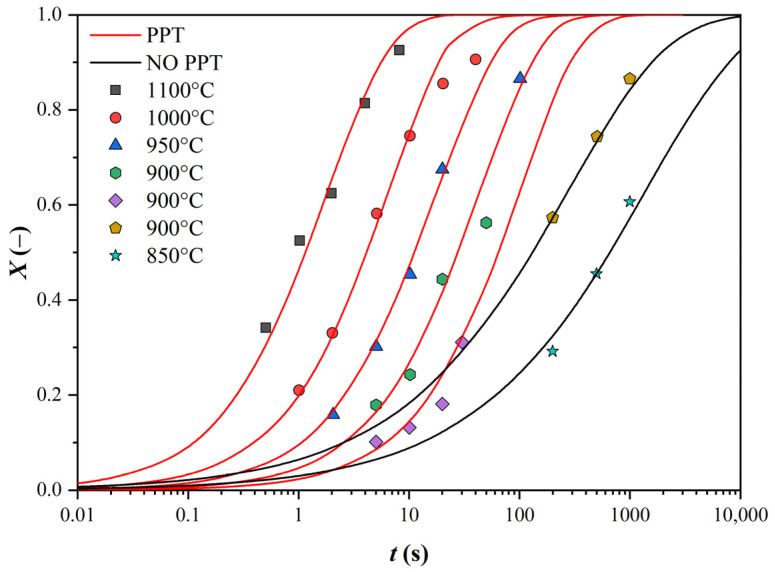
Plotting the experimental data with S-curves of the original data (see Figure 1).

**Figure 3 materials-16-00288-f003:**
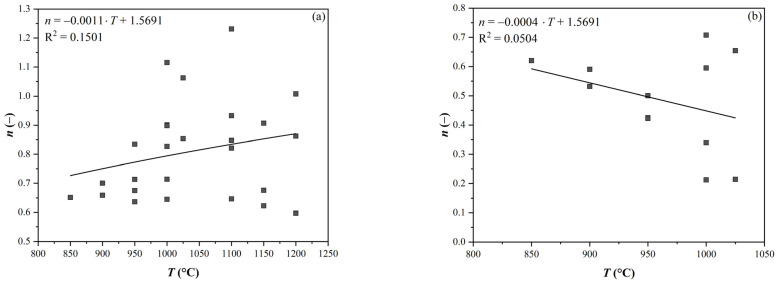
Effect of temperature on the *n* value: (**a**) without SIP and (**b**) with SIP.

**Figure 4 materials-16-00288-f004:**
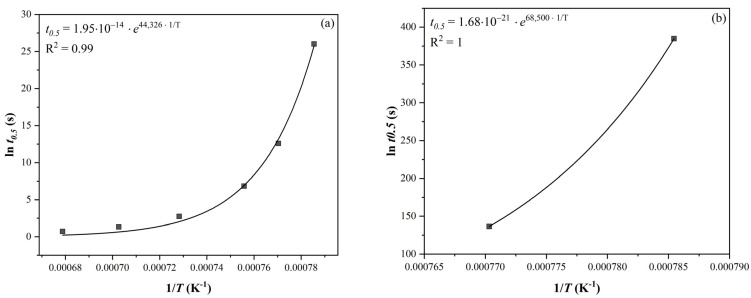
Dependence of *t*_0.5_ on the reciprocal thermodynamic temperature: (**a**) without SIP and (**b**) with SIP (steel N4, *e* = 0.35).

**Figure 5 materials-16-00288-f005:**
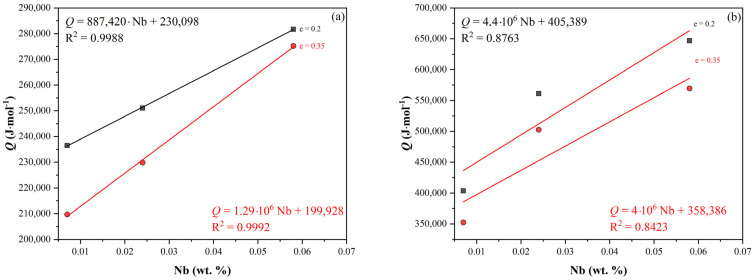
Dependence of *Q* on the strain and Nb content (without SIP): (**a**) without SIP and (**b**) with SIP.

**Figure 6 materials-16-00288-f006:**
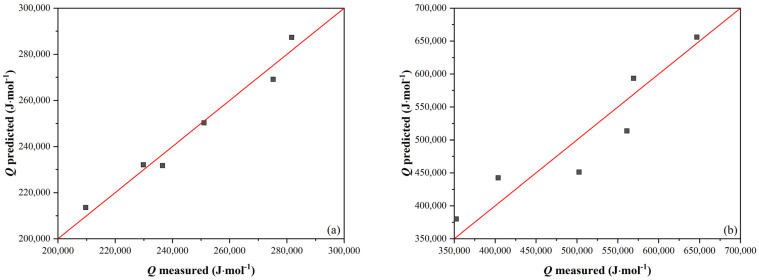
Comparison of measured and calculated values: (**a**) without SIP according to Equation (1) and (**b**) with SIP according to Equation (2).

**Figure 7 materials-16-00288-f007:**
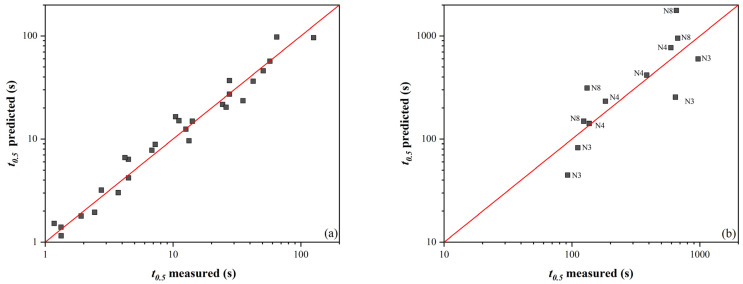
Comparison of measured and calculated *t*_0.5_ values: (**a**) without SIP according to Equation (3), and (**b**) with SIP according to Equation (4) (data labels here represent steel designations.)

**Figure 8 materials-16-00288-f008:**
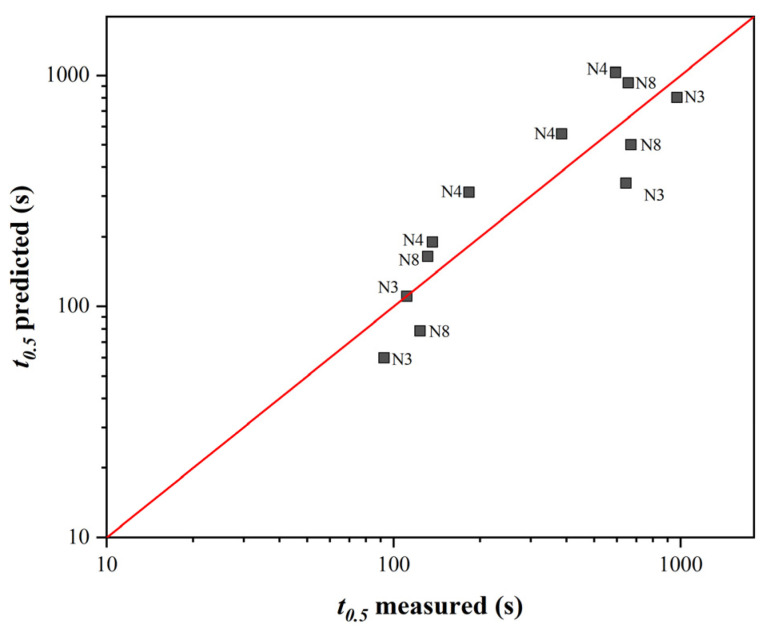
Comparison of measured and calculated *t*_0.5_ values with SIP according to Equation (5) (data labels here represent steel designations).

**Figure 9 materials-16-00288-f009:**
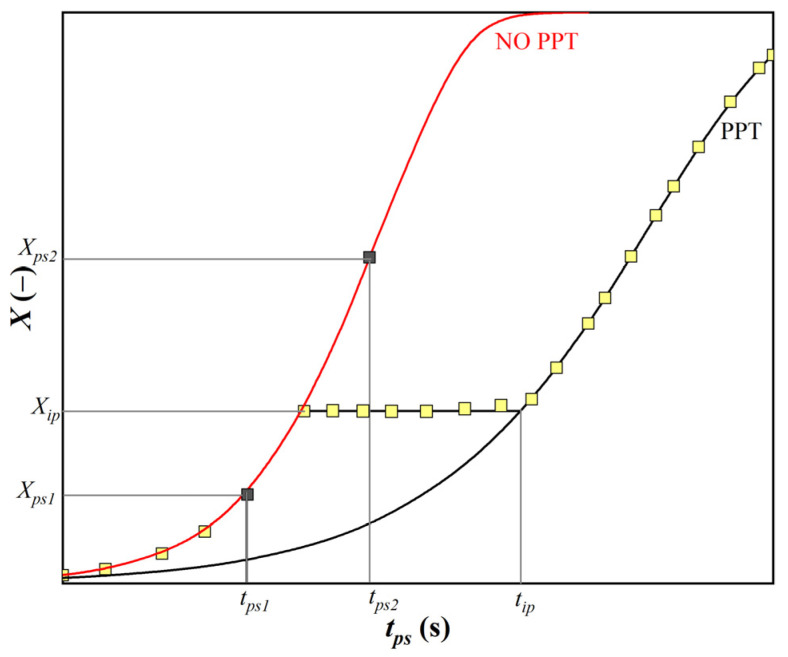
Possible situations that may occur in the collection in which the conditions for the start of SIP are met.

**Figure 10 materials-16-00288-f010:**
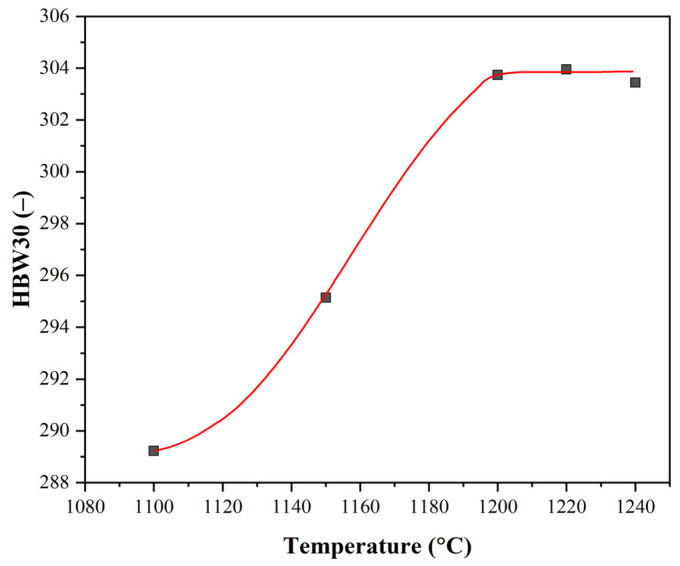
Average hardness of samples after quenching and tempering as a function of heating temperature.

**Figure 11 materials-16-00288-f011:**
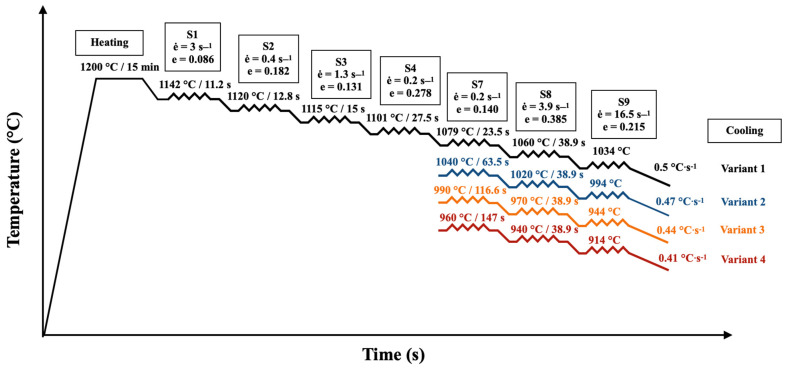
Schematic illustration of PSCTs (
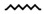
 represents an individual pass).

**Figure 12 materials-16-00288-f012:**
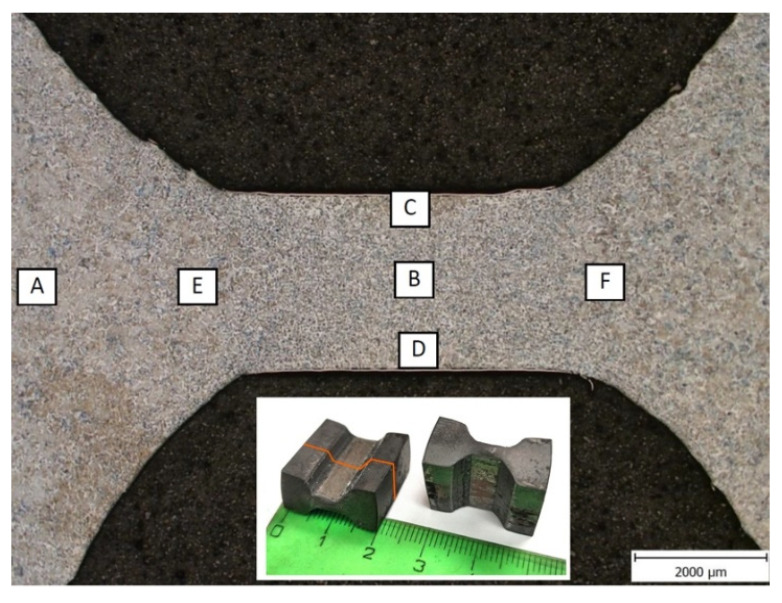
Marking of microstructure documentation sites (sample: variant 1). V1–V4 represent the areas of detailed microstructural analysis.

**Figure 13 materials-16-00288-f013:**
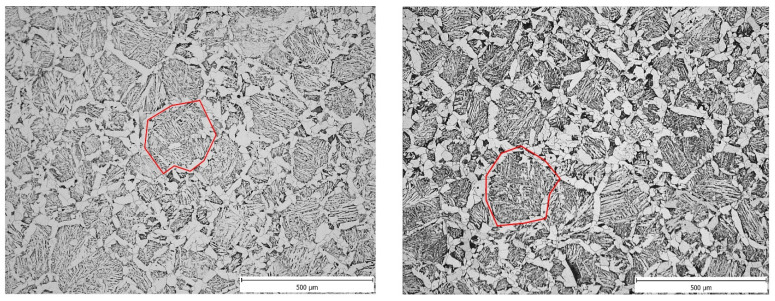
Microstructure consisting of allotriomorphic ferrite, pearlite and acicular ferrite in the undeformed region A: (**left**) sample V1 and (**right**) sample V3. The original austenitic grain boundary is marked in red.

**Figure 14 materials-16-00288-f014:**
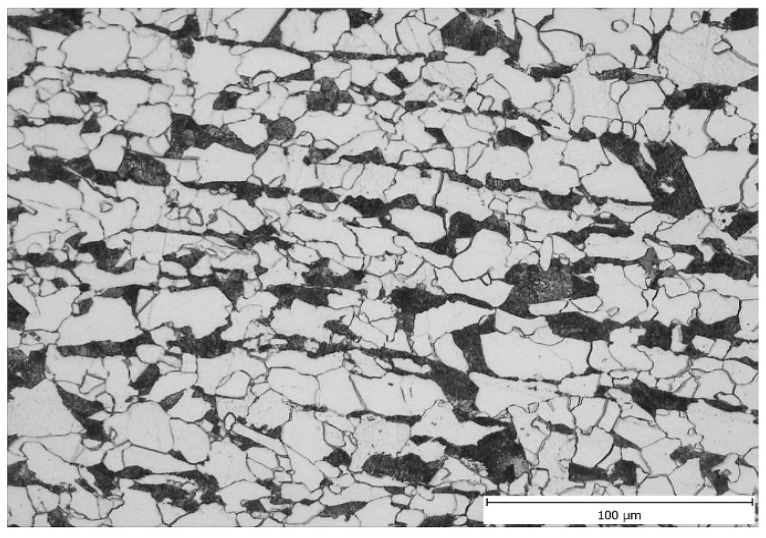
Ferritic–perlitic linear microstructure in the centre (sample V4).

**Figure 15 materials-16-00288-f015:**
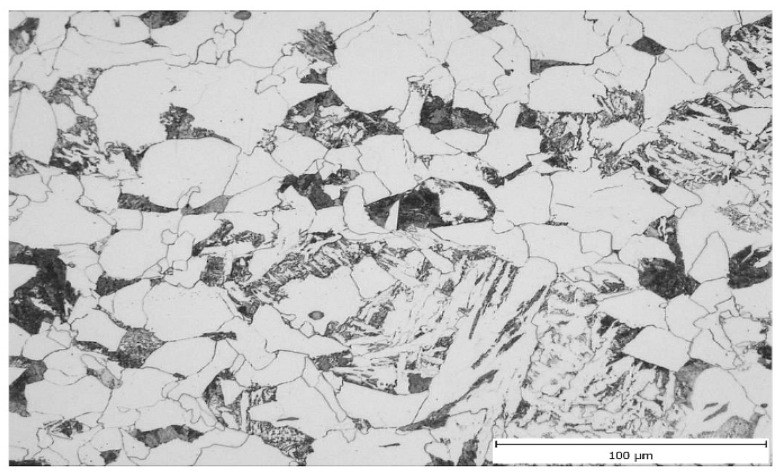
Microstructure consisting of allotriomorphic ferrite, pearlite and acicular ferrite in the centre (sample V1).

**Figure 16 materials-16-00288-f016:**
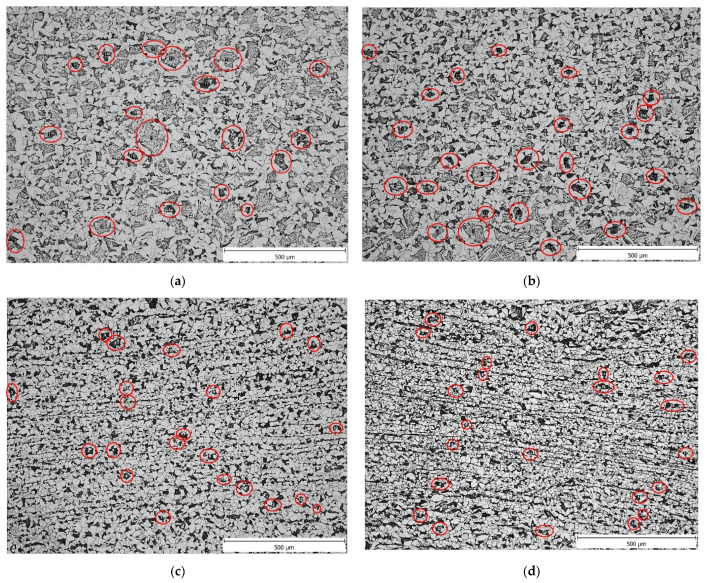
Comparison of the resulting microstructure in the centre of the samples for all variants. V1 and V2—a mixture of allotriomorphic ferrite, acicular ferrite and pearlite; V3 and V4—a mixture of allotriomorphic ferrite and pearlite. The red ellipses indicate the maximum size of the original austenitic grain. (**a**) Variant 1, region B, deformed part; (**b**) variant 2, region B, deformed part; (**c**) variant 3, region B, deformed part; and (**d**) variant 4, region B, deformed part.

**Figure 17 materials-16-00288-f017:**
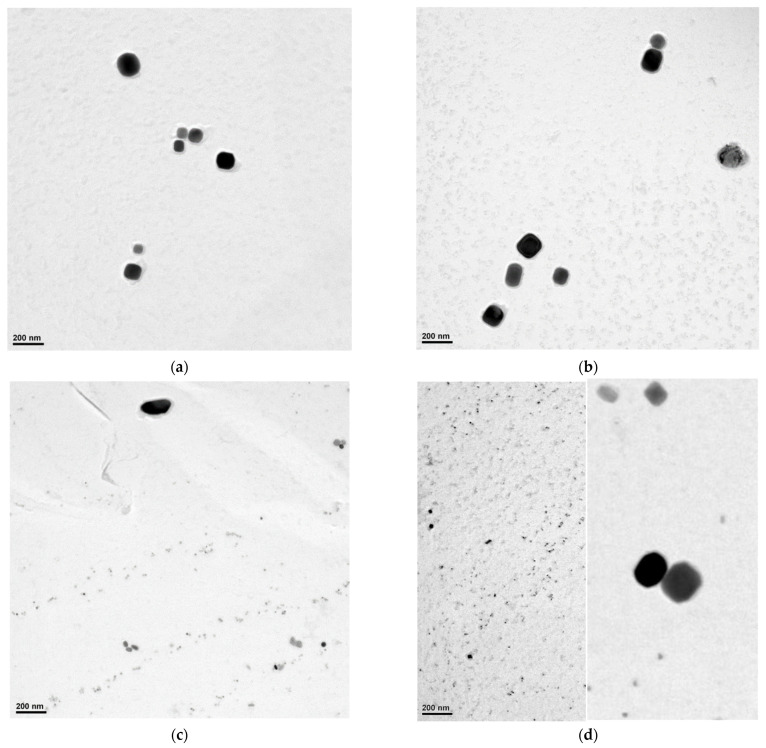
Summary of all precipitate images for each variant V1 to V4. Images were selected that showed examples of precipitates of all observed sizes. For sample V4, it was not possible to capture different-size precipitates in one image; therefore, two images with the same scale are presented. (**a**) V1, (**b**) V2, (**c**) V3, and (**d**) V4.

**Figure 18 materials-16-00288-f018:**
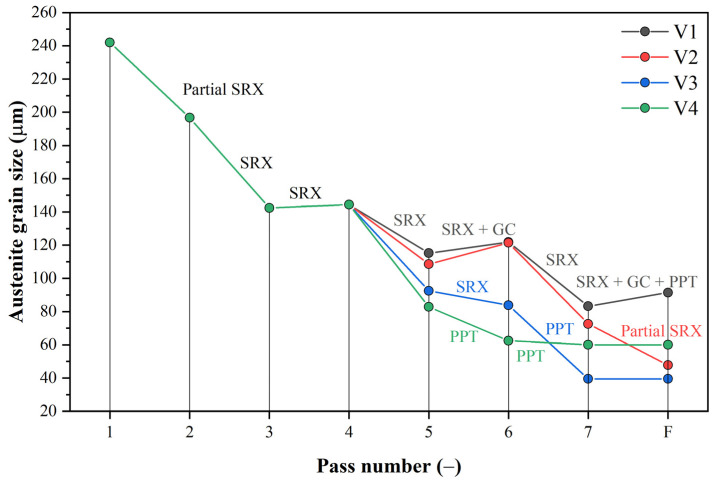
Grain size evolution for all PSCT variants calculated using the modified model.

**Figure 19 materials-16-00288-f019:**
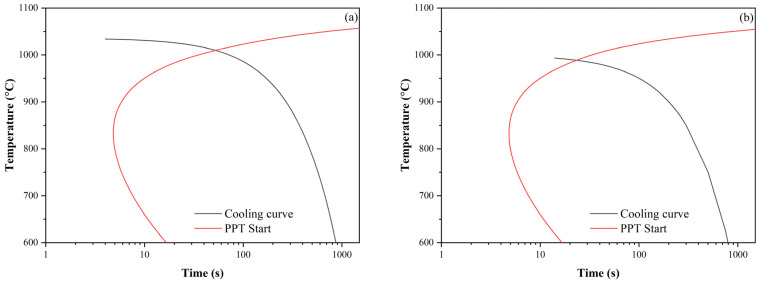
Interaction of cooling curves and precipitation onset curve: (**a**) variant 1 and (**b**) variant 2.

**Figure 20 materials-16-00288-f020:**
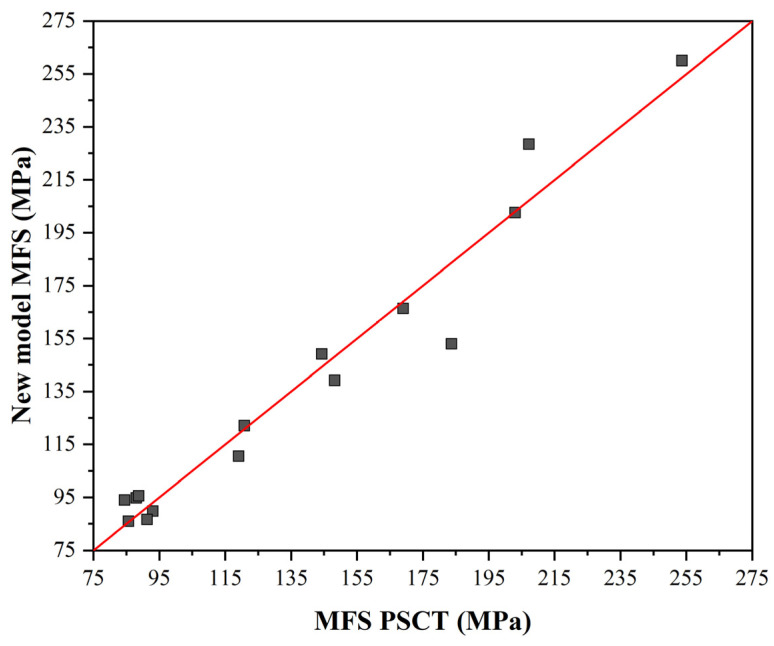
Comparison of measured and calculated MFS values according to Equation (10).

**Figure 21 materials-16-00288-f021:**
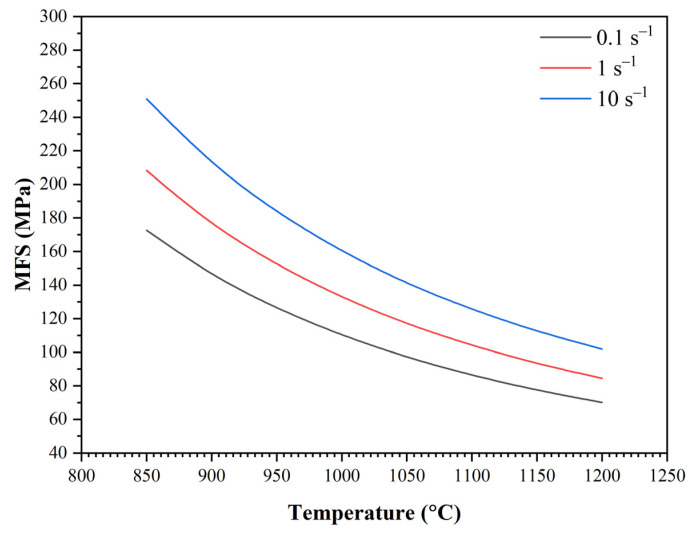
Dependence of MFS on the temperature and strain rate calculated according to Equation (10).

**Figure 22 materials-16-00288-f022:**
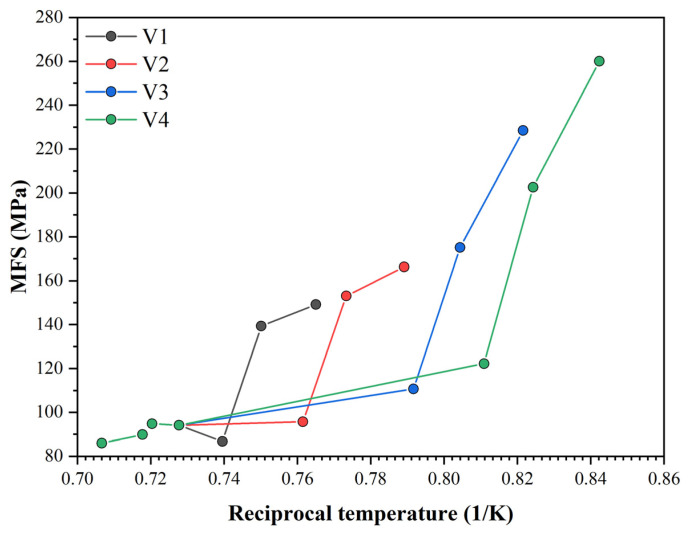
MFS dependencies on reciprocal temperature for each simulation variant.

**Figure 23 materials-16-00288-f023:**
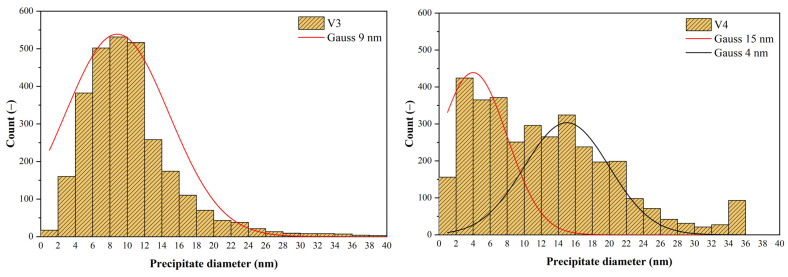
Histograms showing the distribution of precipitates below 40 nm in the samples for V3 (**left**) and V4 (**right**).

**Figure 24 materials-16-00288-f024:**
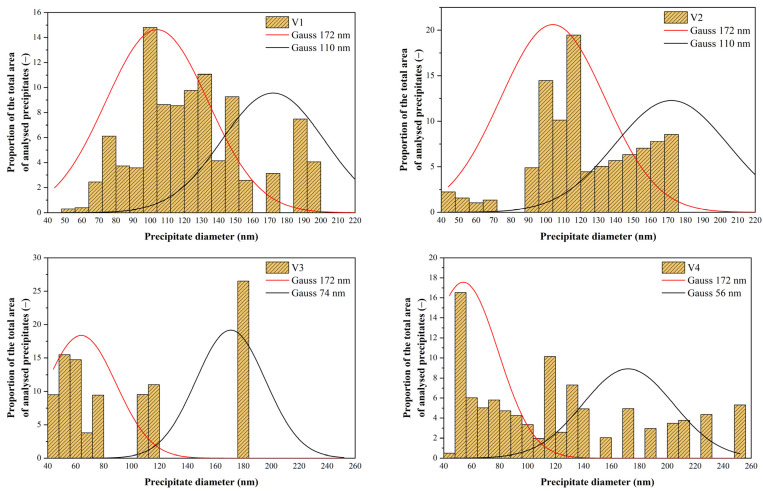
Modified histograms showing the distribution of precipitates above 40 nm in the samples for all variants.

**Figure 25 materials-16-00288-f025:**
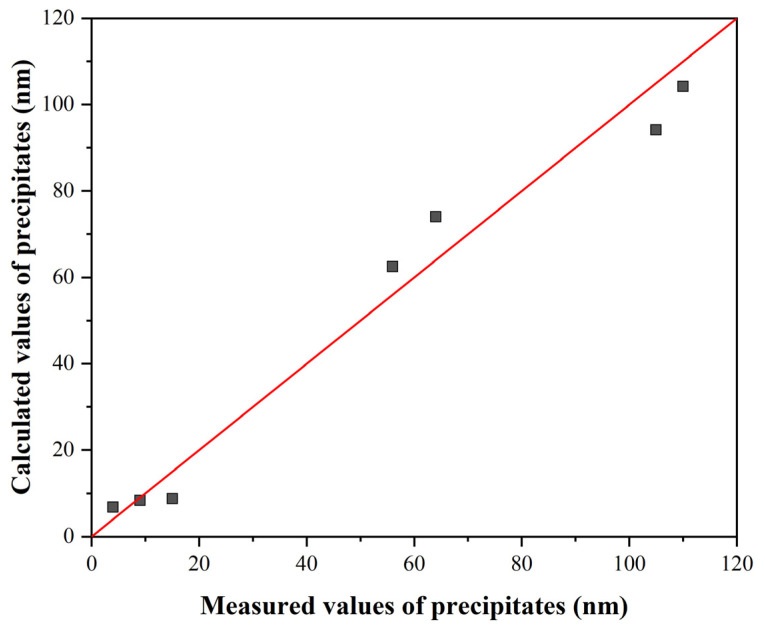
Comparison of measured and calculated values of precipitate size according to Equation (11).

**Figure 26 materials-16-00288-f026:**
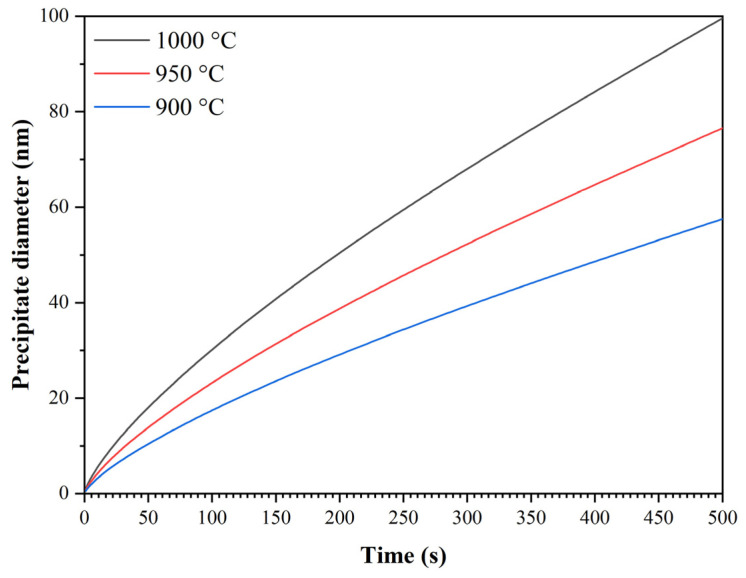
Dependence of precipitate size on the time and temperature of precipitation calculated according to Equation (11).

**Table 1 materials-16-00288-t001:** Chemical composition of steels used for model modification in wt.%.

Steel	C	Si	Mn	P	S	Al	N	Nb
N3	0.21	0.18	1.08	0.023	0.014	0.007	0.0058	0.024
N4	0.21	0.19	1.14	0.023	0.015	0.008	0.0061	0.058
N8	0.2	0.2	1.0	0.024	0.013	0.006	0.0056	0.007

**Table 2 materials-16-00288-t002:** Values of the coefficient *n* in the Avrami equation.

*n* (–)		No SIP									SIP				
Steel	Strain (–)	1200	1150	1100	1050	1025	1000	950	900	850	1025	1000	950	900	850
N3	0.2		0.68		0.86		0.71	0.71				0.34	0.42		
	0.35	0.60	0.62	0.82	0.87		0.90	0.83				0.21	0.50		
N4	0.2	1.01		1.23	1.05	1.06	1.12				0.65	0.71			
	0.35	0.86	0.91	0.93	0.89	0.85	0.90				0.21	0.60			
N8	0.2			0.85			0.83	0.67	0.70				0.42	0.59	
	0.35			0.65			0.65	0.64	0.66	0.65				0.53	0.62
	Mean	0.82	0.74	0.90	0.92	0.96	0.85	0,71	0.68	0.65	0.43	0.46	0.45	0.56	0.62
	Median	0.86	0.68	0.85	0.88	0.96	0.86	0,69	0.68	0.65	0.43	0.47	0.42	0.56	0.62
	Mean	0.82									0.48				
	Median	0.84									0.52				

**Table 3 materials-16-00288-t003:** Values of *t*_0.5_ in the JMAK equation.

*t*_0.5_ (s)			No SIP									SIP				
Steel	*D* (µm)	e (-)	1200	1150	1100	1050	1025	1000	950	900	850	1025	1000	950	900	850
N3	210	0.2		1.91		7.26		24.45	57.25				111.09	970.42		
		0.35	0.77	1.33	2.44	4.49		13.34	35.29				92.56	644.69		
N4	190	0.2	1.33		4.47	10.46	27.66	50.76				183.03	593.68			
		0.35	0.72	1.33	2.75	6.84	12.60	26.01				136.56	384.77			
N8	140	0.2			3.73			14.17	42.33	126.00				123.68	671.66	
		−0.3			1.18			4.21	11.12	27.63	64.85				131.37	656.52

**Table 4 materials-16-00288-t004:** Values of *Q* and *K* in the equation to describe *t*_0.5._

Steel	Strain (-)	No SIP		SIP	
		*Q* (J·mol^−1^)	*K* (-)	*Q* (J·mol^−1^)	*K* (-)
N3		251048	1.08 × 10^−9^	561218	1.05 × 10^−21^
N4	0.2	281684	1.11 × 10^−10^	646761	1.73 × 10^−24^
N8		236541	3.37 × 10^−9^	403719	7.09 × 10^−26^
N3		229855	4.76 × 10^−9^	502583	2.22 × 10^−19^
N4	0.35	275 192	1.09 × 10^−10^	569336	1.68 × 10^−21^
N8		209681	1.19 × 10^−8^	352517	2.64 × 10^−14^

**Table 5 materials-16-00288-t005:** Chemical composition of the selected HSLA steel in wt.%.

C	Mn	Si	P	S	Al	Nb	N	Ti
0.161	1.38	0.178	0.019	0.011	0.035	0.033	0.0044	0.001

**Table 6 materials-16-00288-t006:** Average austenitic grain size values at heating temperature.

Variant	*D* (μm)
1	229
2	258
3	240
4	241

**Table 7 materials-16-00288-t007:** Average austenitic grain size values before phase transformation and average microhardness values.

Variant	*D* (μm)	HV 0.3 (-)
1	96	168
2	72	183
3	50	169
4	46	162

**Table 8 materials-16-00288-t008:** Results of semi-quantitative EDX analysis of the NbX phase (wt.%).

No. of Analysis	Ti	Cr	Fe	Nb
1	0.9	1.6	1.8	96.6
2	0.6	1.6	2.4	96.0
3	0.8	1.4	1.9	96.4
Mean	0.77	1.53	2.03	96.33
Std. deviation	0.153	0.115	0.321	0.306

## Data Availability

Not applicable.

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
