# Peer review of "Analysis of the Microstructure Development of Nb-Microalloyed Steel during Rolling on a Heavy-Section Mill"

_materials, 2022, doi:10.3390/ma16010288_

Round 1

Reviewer 1 Report

Dear authors,

You did an interesting work but the presentation of it does not satisfy completely. My remarks are as follows:

Please revise Abstract by adding the one or two sentences about the motivation for this work. Also, emphasize the aim and importance of the studied problem as well as novelty. Further, list the used methods.

Please describe how values of the coefficient n in the Avrami equation (Table 2) were obtained.

In discussion you should to critically discuss the observations from this work with existing literature.

Used literature is adequate but not really fresh/new. Among 33 references only 2 fall in the last 5 years. Please check and use new references focused on your work. Namely, at least one fourth of all the references could be from 2017 or later years.

The conclusion should be more concise. Also, consider using bullet points.

Best regards

Author Response

Dear reviewer,

Please find the answers to all reviewers questions in the attached document.

Best Regards 

Michal Sauer

Reviewer 2 Report

1.     The purpose of this study is not clear from the “introduction” section. And some experimental processes appear in the “Results and Discussion” section. Therefore, the authors should adjust the structure of this paper.

2.     In the “Microstructure development model” section, the authors mean that the mathematical model in this paper is cited from the published papers? Is the chemical composition of steels in this paper same with the published papers? It seems that the mathematical model is huge influenced by the chemical composition and element.

3.      Line 78: What is the relationship between the austenitic grain evolution and recrystallization and precipitation.

4.     What is the main difference of the microstructure in V1-V4 from Figure 12-15.

5.     Line 287: Do the experimental results consistent with the mathematical model?

6.     Line 86: Figure 1 is blurred.

7.     Line 413: Figure 26 missed.

8.     Line 418: The language in the “Discussion” section should be concise.

9.     What is the advantage of the mathematical model in this paper compared with the published data.

Author Response

Dear reviewer,

I have tried to include your comments as much as possible. Please see the changes in the attached document.

Best Regards

Michal Sauer

Reviewer 3 Report

This paper mainly developed a modified model to describe the microstructure evolution of Nb-microalloyed steel during rolling process. By using the model, the precipitates with different sizes within HSLA steel were simulated and the topography variations of the precipitates under heating and cooling conditions were clarified. Some results are interesting; however, this paper is not prepared well, I would like to reject it. Please carefully double check and improve it. And hereinafter the main comments.

 (1) In abstract, the description seems to have no relation with the title of the paper. The abstract mainly concerns the modified model without research object, while the title focuses on the steel microstructure. In addition, “The resulting grain size for all four modes was consistent with the operating results”, in this sentence, what’s meaning of ‘four models’? Only model was modified, or four models were modified?

(2) In section of conclusions, this is not the ‘conclusion’. It’s a description or some analysis on typical figures or tables.

(3) In Introduction, the literature is reviewed for rolling surface treatment, but the abstract is for the model to predict structure evolution. Therefore, the literature review cannot give the state-of-the-art of present study.

(4) The quality of Fig.1 is poor and hardly shows the variation of two-stage softening.

(5) In table 2, what does it mean for PPT?

(6) The authors claimed that ‘The conformity between the measured and calculated values for t0.5 without PPT is very good, but in the case of t0.5 with PPT, the scatter of the data around the mean line is visibly worse’, this means the fitted lines without PPT is good, but R2 with PPT is smaller (0.8423) than that without PPT (0.9992). Please give further explanation.

(7) In section 3.2, the simulation results were provided. However, Fig.19 has no grain size evolution; in Fig.20, the measured and calculated values cannot be distinguished. Fig.23 and Fig.24 show the distribution of precipitates without pictures to support.

Author Response

(The authors gave the same response as above.)

Reviewer 4 Report

The article is devoted to a very topical topic, namely the study of the effect of hot rolling on the formation of the microstructure and, consequently, mechanical properties. The control of the microstructure and properties during the rolling process makes it possible to increase the level of strength properties and reduce the consumption of metal for the manufacture of products. As a positive side of the reviewed work, one can note the very good quality of the drawings and photographs of the microstructure. However, the article is very poorly structured, which significantly complicates the understanding of the presented research results. I really hope that the authors of the article will treat my comments with understanding and bring the article to the level of an article that can be published in the rating journal Materials.

1. The Reference contains 33 publications, of which more than 50% were published more than 20 years ago. There are no publications with dates 2021...2022. Indeed, studies on rolling are now published not so often, but there are such publications.

For example, https://doi.org/10.3390/ma15124057

https://doi.org/10.3390/ma15124116

https://doi.org/10.3390/met11081239.

2. The information in lines 72...80, in my opinion, should be moved to "1. Introduction". The information from [1–19] needs a more detailed analysis. Indicate what was studied in these works.

3. In section 2.1. there is a description of the developed mathematical model. Is this model designed by you or by Medina S.F.? If not by you, then it is better to transfer the description to "1. Introduction". If it was developed by the authors of the peer-reviewed article, then it should be moved to section “3. Results and discussion".

4. In the recommendations for the design of articles for the journal "Materials" description of materials, experiments and methods should be in section “2. Materials and methods”. Now you have it in section “3. Results and discussion”. This is inconvenient for the reader. Bring the structure of the article in line with the recommendations of the journal.

5. At the end of “1. Introduction” it is necessary to formulate the goals of your research and indicate how they differ from those previously performed by other researchers.

6. The conclusions of the study should be formulated concisely and in accordance with the purpose.

7. For what products is the investigated steel used? What properties should it have? This information should be added to "1. Introduction".

I really hope that the authors will rework the article and publish it.

Author Response

(The authors gave the same response as above.)

Round 2

Reviewer 3 Report

Firstly, the authors claimed that a new model was developed in introduction, however, in the conclusion part, it became a modified model. Both are not shown in the paper title.

Secondly, I cannot find the response of the comments in the word version.

Thirdly, the conclusion part has no improvement compared with previous version.

Therefore, I would like to reject this paper because I cannot find the attitude that you pay more attention to improve the study.

Author Response

Dear reviewer,

the attached document contains responses to your comments.

Best Regards

Michal Sauer

Reviewer 4 Report

The authors responded to my comments and made corrections to the paper. I recommend the article for publication in this version.

Author Response

Dear reviewer,

thank you for taking the time to review the article.

Best Regards

Michal Sauer